# Diagnostic performance of capillary and venous blood samples in the detection of *Loa loa* and *Mansonella perstans* microfilaraemia using light microscopy

**Johannes Mischlinger**[1,2], **Rella Zoleko Manego**[1,2,3], **Ghyslain Mombo-Ngoma**[1,2,3], **Dorothea Ekoka Mbassi**[1,2], **Nina Hackbarth**[1,2], **Franck-Aurelien Ekoka Mbassi**[1,3], **Saskia Dede Davi**[1,2], **Ruth Kreuzmair**[3], **Luzia Veletzky**[1,2], **Jennifer Hergeth**[3], **Wilfrid Nzebe Ndoumba**[3], **Paul Pitzinger**[4], **Mirjam Groger**[1,2], **Pierre Blaise Matsiegui**[5], **Ayôla Akim Adegnika**[3,6,7], **Selidji Todagbe Agnandji**[3,6], **Bertrand Lell**[3,8], **Michael Ramharter**[1,2]*

1 Department of Tropical Medicine, Bernhard Nocht Institute for Tropical Medicine & I. Dep. of Medicine University Medical Center Hamburg-Eppendorf, Hamburg, Germany, 2 German Centre for Infection Research (DZIF), partner site Hamburg-Luebeck-Borstel, Hamburg, Germany, 3 Centre de Recherches Médicales de Lambaréné, Lambaréné, Gabon, 4 Robert Koch-Institute, Berlin, Germany, 5 Centre de Recherches Médicales de la Ngounié (CRMN), Fougamou, Gabon, 6 Institut für Tropenmedizin, Universität Tübingen, Tübingen, Germany, 7 German Center for Infection Research (DZIF), Partner Site Tübingen, Germany, 8 Department of Medicine I, Division of Infectious Diseases and Tropical Medicine, Medical University of Vienna, Vienna, Austria

* ramharter@bnitm.de

## Abstract

### Background

*Loa loa* and *Mansonella perstans*–the causative agents of loiasis and mansonellosis—are vector-borne filarial parasites co-endemic in sub-Saharan Africa. Diagnosis of both infections is usually established by microscopic analysis of blood samples. It was recently established that the odds for detecting *Plasmodium spp.* is higher in capillary (CAP) blood than in venous (VEN) blood. In analogy to this finding this analysis evaluates potential differences in microfilaraemia of *L. loa* and *M. perstans* in samples of CAP and VEN blood.

### Methods

Recruitment took place between 2015 and 2019 at the CERMEL in Lambaréné, Gabon and its surrounding villages. Persons of all ages presenting to diagnostic services of the research center around noon were invited to participate in the study. A thick smear of each 10 microliters of CAP and VEN blood was prepared and analysed by a minimum of two independent microscopists. Differences of log2-transformed CAP and VEN microfilaraemia were computed and expressed as percentages. Furthermore, odds ratios for paired data were computed to quantify the odds to detect microfilariae in CAP blood versus in VEN blood.

**Data Availability Statement:** All relevant data are within the manuscript and its Supporting Information files.

**Funding:** The author(s) received no specific funding for this work.

**Competing interests:** The authors have declared that no competing interests exist.

## Results

A total of 713 participants were recruited among whom 52% were below 30 years of age, 27% between 30–59 years of age and 21% above 60 years of age. Male-female ratio was 0.84. Among 152 participants with microscopically-confirmed *L. loa* infection median (IQR) microfilaraemia was 3,650 (275–11,100) per milliliter blood in CAP blood and 2,775 (200–8,875) in VEN blood (p<0.0001), while among 102 participants with *M. perstans* this was 100 (0–200) and 100 (0–200), respectively (p = 0.44). Differences in linear models amount up to an average of +34.5% (95% CI: +11.0 to +63.0) higher *L. loa* microfilaria quantity in CAP blood versus VEN blood and for *M. perstans* it was on average higher by +24.8% (95% CI: +0.0 to +60.5). Concordantly, the odds for detection of microfilaraemia in CAP samples versus VEN samples was 1.24 (95% CI: 0.65–2.34) and 1.65 (95% CI: 1.0–2.68) for infections with *L. loa* and *M. perstans*, respectively.

## Conclusion

This analysis indicates that average levels of microfilaraemia of *L. loa* are higher in CAP blood samples than in VEN blood samples. This might have implications for treatment algorithms of onchocerciasis and loiasis, in which exact quantification of *L. loa* microfilaraemia is of importance. Furthermore, the odds for detection of *M. perstans* microfilariae was higher in CAP than in VEN blood which may pre-dispose CAP blood for detection of *M. perstans* infection in large epidemiological studies when sampling of large blood quantities is not feasible. No solid evidence for a higher odds of *L. loa* microfilariae detection in CAP blood was revealed, which might be explained by generally high levels of *L. loa* microfilaraemia in CAP and VEN blood above the limit of detection of 100 microfilariae/ml. Yet, it cannot be excluded that the study was underpowered to detect a moderate difference.

## Author summary

Microfilaraemia of *Loa loa* and *Mansonella perstans* was investigated by light microscopy in paired thick smears of capillary and venous blood; each sample was prepared using a standardised quantity of 10 microliters of blood and analysed by a minimum of two independent microscopists. Microfilaraemia was on average +34.5% (95% CI: +11.0 to +63.0) higher in capillary than in venous blood samples for *L. loa* and +24.8% (95% CI: +0.0 to +60.5) for *M. perstans*. This might have implications for treatment algorithms of onchocerciasis and loiasis, in which exact quantification of *L. loa* microfilaraemia is of importance. Furthermore, the odds for detection of *M. perstans* microfilariae was 65% higher in capillary than in venous blood which may pre-dispose capillary blood for detection of *M. perstans* infection in large epidemiological studies when sampling of large blood quantities is not feasible. No solid evidence for a higher odds of *L. loa* microfilariae detection in capillary blood was revealed, which might be explained by generally high levels of *L. loa* microfilaraemia in capillary and venous blood.

## Background

*Loa loa* and *Mansonella perstans* are filarial worms causing loiasis and mansonellosis, respectively [1,2]. They are transmitted by insect vectors, flies of the genus *Chrysops* (for *L. loa*) and midges of

the genus *Culicoides* (for *M. perstans*). Human-to-insect infection occurs by transmission of microfilariae during a blood meal taken by insect vectors. While occurrence of *L. loa* is geographically confined to forested regions of Western and Central Africa *M. perstans* is also reported from other global regions such as the Americas [1,2]. To date, research on both infections is neglected and only in recent years studies have demonstrated that loiasis is associated with higher population morbidity and mortality [3,4]. Regarding mansonellosis little is known on its impact on population morbidity and mortality, however, this might be explained by the relatively lesser attention that it receives in comparison to other tropical diseases. Diagnosis of the infection is established by detection of parasites in biological specimen [5–7]. Most commonly microfilariae are microscopically determined in blood samples. While for *L. loa* a circadian periodicity has been described with highest microfilariae densities around noon, no circadian periodicity is known for *M. perstans* [5–8].

In the past century, several studies investigated the density of filarial blood parasites in samples of capillary (CAP) or venous (VEN) blood over a 24-hour cycle to allow studying of potential circadian periodicities. Most of this research was done for lymphatic filariasis, however, to a lesser extent also for infections with *L. loa* and certain species of *Mansonella* [9–18]. Two studies investigated the microfilarial density and microfilaraemia prevalence, respectively of *L. Loa* and *M. perstans* infections in paired samples of CAP and VEN blood [17,18]. Such information is not merely of academic interest, as higher parasite densities in one blood type can facilitate higher chances of detection by diagnostic tools, as was recently demonstrated for malaria [19]. However, in spite of good methodological quality the findings of the first study are based on only two (n = 2) infected individuals [17] and the second study recruited 39 and 97 individuals infected with *L. loa* and *M. perstans*, respectively [18]. Therefore, in order to add more evidence to this neglected field of research we conducted a large diagnostic study investigating the diagnostic performance characteristics of CAP and VEN blood to detect *L. Loa* and *M. perstans* in infected persons. This study quantified the microfilaraemia of *L. loa* and *M. perstans* in samples of CAP and VEN blood and compared the odds of microfilaria detection in both blood types.

## Methods

### Ethics statement

This study was approved by the ethics committee of CERMEL under the number CEI-013/2018. Participants or their legal representatives gave written informed consent before any study related procedures were performed.

### Patients

Recruitment took place between 2015 and 2019 at the Centre de Recherches Médicales de Lambaréné (CERMEL) in Lambaréné, Gabon and the surrounding villages [20,21]. Persons of all ages presenting to diagnostic services at CERMEL around noon were invited to participate in the study. Participants or their legal representatives gave informed consent before any study related procedures were performed.

At the time of writing of the study protocol a similar diagnostic study in the field of malaria was ongoing at CERMEL [19]. Due to the sameness of the target population, as well as, the recruitment and analytical procedures, samples of the diagnostic malaria study were systematically re-analysed for *L. loa* and *M. perstans* and new participants were recruited in parallel.

### Sample size

Sample size calculation showed that 120 participants with proven filarial infection (either in CAP or VEN blood) are required to be able to detect an odds ratio for paired data of 2.5 using

an alpha of 5% and a power of 80%. Concordantly, in order to confirm a microfilaraemia which is 52.3% higher in one blood sample than in another paired blood sample 65 participants with filarial infection are required. Thus, in order to answer both primary objectives of the study 120 participants each with *L. loa* infection and 120 with *M. perstans* infection are required who are positive for the respective filarial infection in either CAP or VEN blood.

## Materials and parasitological analysis

Thick smears of 10 microliters of blood were prepared and stained with 4% Giemsa for 60 minutes. For each participant two thick smears were prepared, one from CAP and one from VEN blood sampled at the same time. Each thick smear underwent microscopic analysis by two independent microscopists and the arithmetic mean of two results was recorded. Another analysis by a third independent microscopist was performed if the ratio of microfilaraemia from the higher to the lower count was greater than 1.5 or if there was a discrepancy in positivity. In such cases the two results that were closer to each other were taken to calculate the arithmetic mean of microfilaraemia. Microscopists were blinded to each other's results.

## Statistical considerations

STATA16 was used for statistical analyses. Depending on the distribution of the data, paired t-tests or Wilcoxon signed rank tests were used for comparisons of microfilaraemia in CAP and VEN blood. Furthermore, microfilaraemia was log2-transformed and VEN microfilaraemia was subtracted from CAP microfilaraemia. Log-transforming of 0-values is mathematically impossible and if done in statistical software missing values are created. Therefore, in order to be able to include discordant pairs (i.e. CAP+ & VEN- and CAP- & VEN+) a common strategy from the field of microbiology was applied: A value of half the lower limit of detection (i.e. 0.5 microfilariae/microliter blood in a microscopic sample of 10 microliters of blood) was assigned to the negative sample in a discordant pair [19,22]. This strategy is described to increase statistical power and believed to protect from selection bias in the case of this given study. Differences created from two log-transformed continuous variables constitute a ratio after anti-log operations; such ratios allow to visualise the excess microfilaraemia in CAP blood relative to the corresponding VEN microfilaraemia expressed in percent [23,24]. Pearson's r and intraclass correlation coefficients were computed to quantify correlation and reproducibility, respectively of CAP and VEN microfilaraemia.

Contingency tables were created to visualise the distribution of individual prevalence of CAP and VEN microfilaraemia. Odds ratios for paired data were computed to assess the odds of microfilaraemia detection in CAP versus VEN blood samples. McNemar test was used for hypothesis testing of paired proportions. Furthermore, diagnostic sensitivity of CAP and VEN samples was computed by using a light microscopy gold standard. An individual was regarded positive for microfilariae if either a CAP or a VEN sample was positive in microscopy. As such an approach does not yield any false positives, values for specificity (and subsequently any other performance characteristic except sensitivity) are biased and were therefore not computed. Diagnostic sensitivity was computed for the overall study samples and for a sub-population of individuals with a low-level microfilaraemia (i.e. <200 microfilariae/mL blood).

## Results

Between 2015 and 2019 a total of 713 participants were recruited (Table 1). About a quarter (169/710) was below 10 years of age, 17% (120/710) in age group 10–19 years and approximately 10% in every subsequent 10-year age band, ending with 13% (93/710) in the age group of 70 years and older. Age was unknown for three adult participants. 54% (386/711) were

**Table 1. Baseline characteristics.**

| Characteristics | Total (N = 713) |
|---|---|
| | n (column %) |
| Age (n = 710*) | |
| Below 10 years | 169 (23.8%) |
| 10 to 19 years | 120 (16.9%) |
| 20 to 29 years | 79 (11.1%) |
| 30 to 39 years | 56 (7.9%) |
| 40 to 49 years | 69 (9.7%) |
| 50 to 59 years | 67 (9.4%) |
| 60 to 69 years | 57 (8.0%) |
| 70 years and above | 93 (13.1%) |
| Sex (n = 711*) | |
| Female | 386 (54%) |
| Male | 325 (46%) |

*Age or date of birth could not be reported by 3 adult participants and sex was unknown for 2 participants

female and 46% (325/711) were male while sex was unknown for two participants. Among the 713 included participants 152 had a microscopically-confirmed infection with *L. loa* and 102 with *M. perstans*. Recruitment was stopped in 2019 due to logistical reasons despite of only having completed recruitment of 102 instead of the target sample of 120 individuals with confirmed *M. perstans* infection.

For individuals with *L. loa* infection median (IQR) CAP microfilaraemia was 3,650 (275–11,100) per milliliter blood and VEN microfilaraemia was 2,775 (200–8,875) with strong evidence for a true CAP-VEN difference in the target population (p<0.0001) (Table 2). This corresponds to an average of +34.5% (95% CI: +11.0 to +63.0) higher microfilaria quantity in CAP blood compared with the quantity measured in VEN blood (p = 0.0027) (Table 2). Among individuals with *M. perstans* infection a ranked comparison of paired CAP and VEN microfilaraemia indicated similar results with the median (IQR) being identical at 100 (0–200) (p = 0.44). However, borderline evidence for a true CAP-VEN difference in favour of higher microfilaraemia in CAP blood than in VEN blood was revealed in parametric analysis: on average microfilaraemia was 24.8% (95% CI: +0.0 to +60.5) higher in CAP samples compared with VEN samples (p = 0.08).

Correlation between CAP and VEN microfilaraemia was high as demonstrated by Pearson's r of 0.94 and 0.90 for infections with *L. loa* and *M. perstans*, respectively (Table 3). Also, reproducibility of CAP microfilaraemia in VEN microfilaraemia was excellent for both *L. loa* and

**Table 2. Median and mean microfilaraemia in capillary (CAP) and venous (VEN) blood samples.**

| | n | Median microfilaraemia (IQR) | | | Difference of CAP microfilaraemia—VEN microfilaraemia | | |
|---|---|---|---|---|---|---|---|
| | | Capillary (CAP) blood | Venous (VEN) blood | p-value* | Difference log2(CAP)—log2 (VEN) (95% CI) | Excess microfilaraemia in CAP relative to VEN blood (% [95% CI]) | p-value** |
| *Loa loa* | 152 | 3,650 (275 to 11,100) | 2,775 (200 to 8,875) | < 0.0001 | +0.43 (+0.15 to +0.70) | +34.5% (+11.0 to +63.0) | 0.0027 |
| *Mansonella perstans* | 102 | 100 (0 to 200) | 100 (0 to 200) | 0.44 | +0.32 (-0.04 to +0.68) | +24.8% (0.0 to +60.5) | 0.08 |

*Wilcoxon signed-rank test

**t-test; N.B.: Unit: Microfilariae per milliliter blood

**Table 3. Correlation between capillary (CAP) and venous (VEN) microfilaraemia and reproducibility of CAP microfilaraemia in VEN blood.**

| | n | Correlation between CAP microfilaraemia and VEN microfilaraemia | Reproducibility of CAP microfilaraemia in VEN microfilaraemia | |
| --- | --- | --- | --- | --- |
| | | Pearson's r | Intraclass correlation coefficient (95% CI) | p-value* |
| *Loa loa* | 152 | 0.94 | 93.4% (89.4 to 95.8) | < 0.0001 |
| *Mansonella perstans* | 102 | 0.90 | 94.7% (89.3 to 97.4) | < 0.0001 |

*F-test

*M. perstans* infection with intraclass correlation coefficients of 93.4% (p<0.0001) and 94.7% (p<0.0001), respectively.

Contingency tables demonstrate that the number of observations was higher in the discordant pair of 'CAP+ & VEN-' than in the pair of 'CAP- & VEN+' both for infections with *L. loa* and *M. perstans* (Table 4). This indicates that detection of microfilaraemia was more common in CAP than in VEN blood samples. However, while hypothesis tests demonstrate no evidence in support of this finding for *L. loa* infection ('CAP+ & VEN-': 21 and 'CAP- & VEN+': 17; p = 0.52) they provide such evidence for *M. perstans* infection ('CAP+ & VEN-': 43 versus 'CAP- & VEN+': 26; p = 0.041). Concordantly, the odds for detection of microfilaraemia in CAP samples versus that in VEN samples was 1.24 (95% CI: 0.65–2.34) and 1.65 (95% CI: 1.0–2.68) for infections with *L. loa* and *M. perstans*, respectively.

As evaluated against the microscopy CAP-VEN gold-standard the diagnostic sensitivity was higher in CAP samples than in VEN samples (Tables 5 and 6). CAP sensitivity was 88.8% and VEN sensitivity 86.2% for *L. loa* infection and 74.5% and 57.8%, respectively for *M. perstans* infection. For participants with a low-level microfilaraemia (i.e. <200 microfilariae/mL) CAP-VEN sensitivity differences became larger, particularly for infection with *M. perstans*: CAP sensitivity was 58.8% and VEN sensitivity was 50.0% for *L. loa* infection and for *M. perstans* infection 66.7% and 47.2%, respectively.

## Discussion

This analysis indicates that microscopically-determined microfilaraemia is higher in CAP blood samples than in VEN blood samples for *L. loa* infection and *M. perstans* infection. Previous studies show concordant evidence in favour of higher *L. loa* and *M. perstans* microfilaraemia in CAP than in VEN blood samples when equal blood volumes were evaluated. In 1950, Kershaw assessed CAP and VEN blood samples of each 50 microliters blood taken from two participants and demonstrated a higher microfilarial blood density in CAP blood than in VEN blood [17]. Later, in 1990 Noireau and Apembet assessed the diagnostic sensitivity of 40

**Table 4. Crosstabulation of capillary (CAP) and venous (VEN) microfilaraemia and odds ratios to quantify the odds to detect a microfilaraemia in CAP blood than in VEN blood.**

| | | | | Microscopy CAP | | Odds ratio for paired data (95% CI) | |
| --- | --- | --- | --- | --- | --- | --- | --- |
| N = 713 | | | | n, column % | | | |
| | | | | + | - | | p-value* |
| *Loa loa* | Microscopy | VEN | + | 114 | 17 | 1.24 (0.65 to 2.34) | 0.52 |
| | | | - | 21 | 561 | | |
| *Mansonella perstans* | Microscopy | VEN | + | 33 | 26 | 1.65 (1.0 to 2.68) | 0.041 |
| | | | - | 43 | 611 | | |

*McNemar test

**Table 5. Crosstabulation of capillary and venous microfilaraemia versus a light microscopy gold-standard.**

| | | | | Microscopy (CAP or VEN)* | |
|---|---|---|---|---|---|
| **N = 713** | | | | **n, column %** | |
| | | | | **+** | **-** |
| *Loa loa* | Microscopy | Only CAP | + | 135 | 0 |
| | | | - | 17 | 561 |
| | Microscopy | Only VEN | + | 131 | 0 |
| | | | - | 21 | 561 |
| *Mansonella perstans* | Microscopy | Only CAP | + | 76 | 0 |
| | | | - | 26 | 611 |
| | Microscopy | Only VEN | + | 59 | 0 |
| | | | - | 43 | 611 |

* positive = either CAP+ & VEN+; CAP+ & VEN-; CAP- & VEN+

* negative = CAP- & VEN-

microliters of CAP blood samples in 39 and 97 individuals infected with *L. loa* and *M. perstans*, respectively whose infection status was ascertained by light microscopical assessment of 5ml VEN blood lysed with 2% saponin [18]. They indicated that 40 microliters of CAP blood had a sensitivity of 67% (26/39) and 62% (60/97) to detect *L. loa* and *M. perstans*, respectively. However, a direct comparison of diagnostic properties of CAP and VEN blood samples was not possible, as preparation methods and blood quantities were different prior to diagnostic assessment. Applying the exact same diagnostic methods and investigating equal volumes of blood we noted that *L. loa* microfilaraemia was about 33% higher in CAP blood than in VEN blood (p = 0.0027) and about 25% higher for *M. perstans* infection (p = 0.08). Interestingly, while the best model estimate indicates increased odds for *L. loa* microfilariae detection in CAP blood (OR: 1.24) compared with VEN blood, formal hypothesis testing did not reveal any evidence for such an effect in the target population (p = 0.52). Yet, as the applied parameters in our sample size calculation might have been overestimated (i.e. estimated OR: 2.5) there is a possibility that the study was underpowered to detect a true effect. However, in microscopy the probability of detection of blood parasite increases, as the parasite quantity increases and the limit of detection in this study was 100 microfilariae per one milliliter blood. That being said, the median *L. loa* microfilaraemia in our study sample was high both in CAP (3,650 [IQR: 275–11,100]) and in VEN blood (2,775 [IQR: 200–8,875]). Therefore, if the true population medians of CAP and VEN microfilaraemia are truly above the limit of detection in microscopy, then absolute differences in CAP-VEN microfilaraemia might not contribute to a differential odds of microfilaria detection. On the other hand, there was an increased odds for *M. perstans* microfilariae detection in CAP blood compared with that in VEN blood (OR: 1.65; p = 0.041). Such increased odds of microfilaria detection in CAP blood seem plausible as median CAP

**Table 6. Diagnostic sensitivity of capillary and venous blood to detect a microfilaraemia using microscopy.**

| | | Total (N = 713) | | Low-level microfilaraemia* | |
|---|---|---|---|---|---|
| | | **CAP % (95% CI)** | **VEN % (95% CI)** | **CAP % (95% CI)** | **VEN % (95% CI)** |
| *Loa loa* | Sensitivity | 88.8 (86.5 to 91.1) | 86.2 (83.7 to 88.7) | 58.8 (54.9 to 62.8) | 50.0 (46.0 to 54.0) |
| *Mansonella perstans* | Sensitivity | 74.5 (71.3 to 77.7) | 57.8 (54.2 to 61.5) | 66.7 (63.1 to 70.2) | 47.2 (43.5 to 51.0) |

*Defined as <200 microfilariae/milliliter for *Loa loa* (n = 595) and *Mansonella perstans* (n = 683)

and VEN *M. perstans* microfilaraemia was exactly at the microscopic limit of detection and therefore even slightly lower levels microfilaraemia in VEN blood can explain the increased odds for detection of microfilariae in CAP blood (i.e. +25% higher microfilaraemia in CAP blood; p = 0.08). This hypothesis is supported by the increasing CAP-VEN sensitivity difference in a sub-population of participants with a low *M. perstans* microfilaraemia. Additionally, the study might be underpowered to detect an absolute difference in CAP-VEN *M. perstans* microfilaraemia, as average values in our study sample were much lower than estimated in sample size calculations (50% estimated versus 25% observed). Furthermore, although this cannot be supported by data of our study it may also be possible that in a larger sample of participants *L. loa* microfilaraemia might be more often detected in CAP than in VEN blood, given the relatively higher values for microfilaraemia in CAP blood relative to those in VEN blood. However, it is of mention that to the best of our knowledge this is the largest study to date on this topic.

Traditionally, infection with *L. loa* received mainly attention in areas of onchocerciasis co-endemicity, as high *L. loa* microfilaraemia can lead to potentially lethal encephalopathy during administration of ivermectin (i.e. the treatment used in mass drug administration campaigns against onchocerciasis). Consequently, it is recommended that *L. loa* microfilaraemia needs to be below a certain threshold before treatment with ivermectin and other strong microfilaricidal agents can be safely administered. Therefore, as for other tropical pathogens the quantification of *L. loa* microfilaraemia and knowledge of pre-analytical influences such as those by CAP and VEN blood sampling are important [25,26]. For patient safety, it might therefore be beneficial if microfilaraemia levels are below the given target threshold in CAP blood, based on findings of this study, that CAP microfilaraemia levels are on average about 33% higher than those in VEN blood. In the past, the implementation of ivermectin-based mass drug administration for the control of onchocerciasis or lymphatic filariasis has been impaired because of afore-mentioned serious adverse events in persons with a high *L. loa* microfilaraemia (i.e. > 20,000 microfilariae/milliliter blood) [27]. To overcome this treatment gap, a so called 'Test and not treat' strategy was developed which is based on withholding mass drug administration from those people with high levels of *L. loa* microfilariae [28,29]. Such highly microfilaraemic persons are identified by means of rapid testing with a diagnostic tool named LoaScope, which analyses samples of peripheral blood [28,30]. Therefore, in order to ensure safety of ivermectin mass drug administration CAP blood should be used for rapid quantification of *L. loa*. microfilariae with LoaScope. It is likely that CAP blood sampling may already constitute the main blood sampling method for rapid assessments, as it is an easy procedure even under field conditions. However, besides this feasibility aspect this current study adds an additional safety argument in favour of CAP blood sampling.

Generally, studies on antimicrobial chemotherapy often rely on the exact quantification of microbial blood density in study participants. Therefore, findings of this present study should be considered for protocols of studies on the therapy of loiasis and mansonellosis. Additionally, in spite of evidence for a higher average microfilaraemia in CAP blood versus VEN blood regarding infections with *L. loa* and *M. perstans*, highly favourable intraclass correlation coefficients demonstrate that a CAP microfilaraemia is well reproducible in VEN blood and vice versa.

In 1917, Yorke and Blacklock discussed that microfilariae of *Wuchereria bancrofti* are more abundant in CAP than in VEN blood irrespective of varying microfilarial peak densities during a 24-hour cycle. This phenomenon was explained by mechanical obstruction of microfilariae during passage through the small cutaneous capillary blood vessels aiding in the piling up of larvae in the cutaneous vessels [9]. It is likely that this explanation also applies to microfilariae of *L. loa* and *M. perstans* which are similar in size (about 275x7µm and 210x4µm, respectively)

[1,2]. Furthermore, from an evolutionary-biological perspective a high quantity of microfilariae in CAP blood seems plausible, as CAP blood is taken up during a blood meal by the insect vector [31]. Thereby, a high CAP microfilaraemia facilitates favourable chances for transmission of the infection to the insect-vector and might consequently contribute to the survival of the respective parasite species.

It needs to be mentioned, that the probability of microfilaraemia detection by light microscopy does not only potentially depend on the type of blood sample (i.e. CAP or VEN) used for diagnostic purposes, but also on the quantity of the investigated sample volume. This study only investigated 10 microliters of blood, a blood quantity which is common particularly for diagnosis of malaria in clinical and research settings. This constitutes a limitation, as it is possible to extract higher quantities of CAP blood during a given CAP blood sampling session (e.g. 50 microliters). Furthermore, in individual case management when infection by a filarial blood parasite is suspected it is recommended to withdraw venous blood of larger quantities (i.e. several milliliters) for microfilariae detection at different days [7,18]. Yet, it was the objective of this study to investigate CAP-VEN differences in microfilaraemia and a potentially resulting differential odds for microfilaraemia-detection. For that it was important to investigate equal sample volumes of CAP and VEN blood in order to enable a fair comparison. A quantity of 10 microliters was considered as feasible to be assessed in sufficient detail by several microscopists. The applied two-to-three microscopist approach is a methodological strength and ensures exact quantification of microfilaraemia so that measurement errors are considered minimal. Furthermore, it is a strength that all microscopists were blinded to each other's results and received training on the diagnostic features of the two target microfilariae of *L. loa* and *M. perstans*.

## Conclusions

This analysis indicates strong evidence that levels of microfilaraemia of *L. loa* are higher in blood samples of CAP blood than in samples of VEN blood. This might have implications for treatment algorithms of onchocerciasis and loiasis, for which exact knowledge of *L. loa* microfilaraemia is important. Additionally, findings of this study might be beneficial for studies on the treatment of loiasis and mansonellosis. Furthermore, the odds for detection of *M. perstans* microfilariae was higher in CAP than in VEN blood which may pre-dispose CAP blood for detection of *M. perstans* infection in large epidemiological studies when sampling of large blood quantities is not feasible. No solid evidence for a higher odds of *L. loa* microfilariae detection in CAP blood was revealed. This might be explained by the overall high levels of *L. loa* microfilaraemia in both CAP and VEN blood which were generally above the microscopic limit of detection in both blood sources; also, it is possible that the study was underpowered to detect such an effect.

## Supporting information

**S1 Dataset. Underlying data.**
(DTA)

## Author Contributions

**Conceptualization:** Johannes Mischlinger, Michael Ramharter.

**Data curation:** Johannes Mischlinger, Dorothea Ekoka Mbassi, Nina Hackbarth, Saskia Dede Davi.

**Formal analysis:** Johannes Mischlinger.

**Investigation:** Johannes Mischlinger, Rella Zoleko Manego, Ghyslain Mombo-Ngoma, Dorothea Ekoka Mbassi, Nina Hackbarth, Franck-Aurelien Ekoka Mbassi, Saskia Dede Davi, Ruth Kreuzmair, Luzia Veletzky, Jennifer Hergeth, Wilfrid Nzebe Ndoumba, Paul Pitzinger, Mirjam Groger, Pierre Blaise Matsiegui, Ayôla Akim Adegnika, Selidji Todagbe Agnandji, Bertrand Lell.

**Methodology:** Johannes Mischlinger, Michael Ramharter.

**Project administration:** Johannes Mischlinger, Rella Zoleko Manego, Dorothea Ekoka Mbassi, Ruth Kreuzmair.

**Resources:** Johannes Mischlinger, Rella Zoleko Manego, Ghyslain Mombo-Ngoma, Pierre Blaise Matsiegui, Ayôla Akim Adegnika, Selidji Todagbe Agnandji, Bertrand Lell, Michael Ramharter.

**Software:** Johannes Mischlinger.

**Supervision:** Johannes Mischlinger, Rella Zoleko Manego, Michael Ramharter.

**Validation:** Johannes Mischlinger.

**Visualization:** Johannes Mischlinger.

**Writing – original draft:** Johannes Mischlinger, Michael Ramharter.

**Writing – review & editing:** Johannes Mischlinger, Rella Zoleko Manego, Ghyslain Mombo-Ngoma, Dorothea Ekoka Mbassi, Nina Hackbarth, Franck-Aurelien Ekoka Mbassi, Saskia Dede Davi, Ruth Kreuzmair, Luzia Veletzky, Jennifer Hergeth, Wilfrid Nzebe Ndoumba, Paul Pitzinger, Mirjam Groger, Pierre Blaise Matsiegui, Ayôla Akim Adegnika, Selidji Todagbe Agnandji, Bertrand Lell, Michael Ramharter.

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
