## [Decision Letter · Decision Letter 0]

5 May 2021

Dear MD Mischlinger,

Thank you very much for submitting your manuscript "Diagnostic performance of capillary and venous blood samples in the detection of Loa loa and Mansonella perstans microfilaraemia using light microscopy" for consideration at PLOS Neglected Tropical Diseases. As with all papers reviewed by the journal, your manuscript was reviewed by members of the editorial board and by several independent reviewers. In light of the reviews (below this email), we would like to invite the resubmission of a revised version that takes into account the reviewers' comments. 

We cannot make any decision about publication until we have seen the revised manuscript and your response to the reviewers' comments. Your revised manuscript is also likely to be sent to reviewers for further evaluation.

Sincerely,

Andrés F. Henao-Martínez, M.D.

Deputy Editor

Reviewer's Responses to Questions

**Key Review Criteria Required for Acceptance?**

**Methods**

-Are the objectives of the study clearly articulated with a clear testable hypothesis stated?

-Is the study design appropriate to address the stated objectives?

-Is the population clearly described and appropriate for the hypothesis being tested?

-Is the sample size sufficient to ensure adequate power to address the hypothesis being tested?

-Were correct statistical analysis used to support conclusions?

-Are there concerns about ethical or regulatory requirements being met?

Reviewer #1: The objective of the study conducted by Mischlinger et al. was to compare the number of Loa loa and Mansonella perstans microfilariae observed in 10 µL of venous blood and 10 µL of capillary blood. The authors found that the microfilarial densities (MfD) were significantly higher in capillary blood and state that this finding is important because it “might have implications for treatment algorithms of onchocerciasis and loiasis, in which exact quantification of L. loa microfilaraemia is of importance”.

The authors say their study is the first to be conducted to compare Loa and M. perstans MfDs in venous and capillary blood. This is not correct. Kershaw (Ann Trop Med Parasitol 1950) studied this on two subjects (and found that the L. loa and M. perstans MfDs are higher in capillary blood). Similarly, Noireau et al. (J Trop Med Parasitol 1990) compared L. loa and M. perstans MfDs in venous and capillary blood collected from 201 subjects. In addition, the fact that the MfD is higher in capillary blood than in venous blood is also very well known for Wuchereria bancrofti, the cause of lymphatic filariasis, whose microfilariae are similar in size to those of L. loa. It is very strange that the authors did not review the studies comparing different methods used to quantify W. bancrofti MfD from venous and capillary blood. One of the first articles is that of Yorke & Blacklock, Ann Trop Med Parasitol 1917. Over 15 other articles present data on this subject. This review should have been included in the introduction or in the discussion section of the paper.

The authors collected 10 µL of capillary blood and 10 µL of venous blood (without fully explaining how they proceeded for the venous blood). As they write themselves, this quantity is very low but they explain that it "was restricted by the small extractable quantities of CAP blood". This argument does not hold since the volume of blood routinely (and easily) taken by finger prick during filariasis surveys is 50 µL. It would have been more interesting to compare the MfDs in two 50 µL samples, which would have detected more subjects with low microfilaremia.

The statistical methods used to analyse the data are relatively complex. The authors write rightly that the logarithm of 0 does not exist but it is common in the field of filariasis to add 1 to the MfD to be able to calculate geometric means taking into account subjects with negative results (Williams’ mean).

Reviewer #2: The objectives were clearly stated and the study design was appropriate, the study population was also clearly described and appropriate though the adequate sample size was not achieved to ensure adequate power (acknowledged by the researchers)

Ethical statement, should state the ethics committee approval number.

There are problems with clarity especially in the methods and results section for non-specialists. The authors need an English specialist to review that section and correct some grammatical presentations.

Reviewer #3: (No Response)

**Results**

-Does the analysis presented match the analysis plan?

-Are the results clearly and completely presented?

-Are the figures (Tables, Images) of sufficient quality for clarity?

Reviewer #1: The analysis presented match the analysis plan and the results are clearly and completely presented.

Reviewer #2: Yes the analysis presented match the analysis plan but the prose of the results needs to be made clearer. On the other hand, the tables are clearly presented.

Reviewer #3: (No Response)

**Conclusions**

-Are the conclusions supported by the data presented?

-Are the limitations of analysis clearly described?

-Do the authors discuss how these data can be helpful to advance our understanding of the topic under study?

-Is public health relevance addressed?

Reviewer #1: By reading the literature on W. bancrofti mentioned above, the authors could have discussed the hypotheses proposed by the various authors who have worked on the subject to explain the phenomenon of higher MfD in capillary blood. In the manuscript, they just say that in malaria the differences “can be explained by host-parasite receptor interactions”. However, it is clear that the microfilariae are large organisms (unlike Plasmodium, to which the authors refer several times) and that the observed differences are probably at least partly due to the ratios between the size of the parasites and the diameter of blood vessels.

When the authors say that their findings may have implications for the treatment of onchocerciasis and loiasis because it is necessary to know the exact Loa MfD before giving treatment, they overestimate somewhat the interest of their results. Actually, the decisions are made according to the MfD measured by the reference method which is the quantification in capillary blood (which enables to detect at-risk subjects with high MfD). The fact that the density is lower in the venous blood does not really matter.

Reviewer #2: The conclusions are supported by the presented results, and the limitations of analysis clearly described. The authors have discussed adequately the implications of the data presented and its relevnce to public health.

Reviewer #3: (No Response)

**Editorial and Data Presentation Modifications?**

Reviewer #1: Line 83, delete “(L.)” and “(M.)”

Line 85, delete both “spp.”

Line 97: cite Asio et al., Parasitology Research 2009

Line 103: in the list of references, reference 8 is incomplete

Line 140: check reference number

Reviewer #2: The sample size determination should come before materials and parasitological analysis, not after statistical considerations.

Reviewer #3: (No Response)

**Summary and General Comments**

Reviewer #1: This paper is of limited interest. The results confirm those presented in other papers on Loa loa, M. perstans, and W. bancrofti. They are not very original and will not have any implication on the strategies to combat onchocerciasis in loiasis-coendemic areas.

Reviewer #2: The study adds some value to the body of knowledge already known in this area. Identifying that capillary blood samples yield higher levels of parasitaemia has been known for malaria, but not for microfilaria. That is what this study fills in.

The details of the methodology are sufficient to reproduce the study

There are problems with clarity especially in the methods and results section for non-specialists. The authors need an English specialist to review that section and correct some grammatical presentations. Major ones have been highlighted here. The sample size determination should come before materials and parasitological analysis, not after statistical considerations.

• The academic achievements should be removed from the authors names

• Line 117, ‘two thick smears of blood, 10 microliters each were prepared per participants’

• Line 118, ‘one from capillary and one from venous blood at the same time’

• Line 162, ‘120 participants each with L. loa infection and 120 with M. perstans infection are required’

• Ethical statement, should state the ethics committee approval number

• Line 170, participants were

• Line 172, subsequent 10-year age group

• Line 182 remove revealed

• Line 282 largest study to date

• Information in line 303, is better put in the methods section

• Lines 327, 338, the references are incomplete

Reviewer #3: The paper by Mischlinger et al. presents the results of comparing measurements of microfilariae of Loa loa and Mansonella perstans between venous and capillary blood collected from 201 patients.

The paper presents an incomplete bibliography as there are two papers that have already performed this experiment; and a large number of papers on Wuchereria bancrofti. The authors state that this is the first experiment, which is not the case. The authors should complete the bibliography accordingly. Instead, the comparison with malaria is throughout the paper, which is less coherent.

The authors should clarify in which framework this study was done. Indeed, it is obvious that this is an ancillary study and that it was not prepared for this purpose. In particular, this would explain why only 10 μL were collected from both venous and capillaries. It would be good to clarify this, and to put less emphasis on possible technical difficulties, as otherwise readers not used to this disease might think that it is difficult to perform capillary or venous sampling, which is not the case.

Last, it seems that the authors should more discuss about the raison about the observed difference.

Finally, the authors indicate that these results could have implications for control programmes, but this is difficult to relate. The samples used are always capillary, and the thresholds have been defined for this type of sampling, so, in my opinion, these results have no obvious implications.

In the end, although the analyses are well done, the paper may not be of sufficient interest for publication in Plos NTD.

PLOS authors have the option to publish the peer review history of their article (what does this mean?). If published, this will include your full peer review and any attached files.

Reviewer #1: No

Reviewer #2: No

Reviewer #3: No
---

## [Decision Letter · Decision Letter 1]

3 Jul 2021

Dear MD Mischlinger,

We are pleased to inform you that your manuscript 'Diagnostic performance of capillary and venous blood samples in the detection of Loa loa and Mansonella perstans microfilaraemia using light microscopy' has been provisionally accepted for publication in PLOS Neglected Tropical Diseases.

Best regards,

Andrés F. Henao-Martínez, M.D.

Deputy Editor

Andrés Henao-Martínez

Deputy Editor

Reviewer's Responses to Questions

**Key Review Criteria Required for Acceptance?**

**Methods**

-Are the objectives of the study clearly articulated with a clear testable hypothesis stated?

-Is the study design appropriate to address the stated objectives?

-Is the population clearly described and appropriate for the hypothesis being tested?

-Is the sample size sufficient to ensure adequate power to address the hypothesis being tested?

-Were correct statistical analysis used to support conclusions?

-Are there concerns about ethical or regulatory requirements being met?

Reviewer #1: Yes for all questions except the last

No ethical concern

Reviewer #2: (No Response)

**Results**

-Does the analysis presented match the analysis plan?

-Are the results clearly and completely presented?

-Are the figures (Tables, Images) of sufficient quality for clarity?

Reviewer #1: Yes

Reviewer #2: (No Response)

**Conclusions**

-Are the conclusions supported by the data presented?

-Are the limitations of analysis clearly described?

-Do the authors discuss how these data can be helpful to advance our understanding of the topic under study?

-Is public health relevance addressed?

Reviewer #1: Yes

Reviewer #2: (No Response)

**Editorial and Data Presentation Modifications?**

Reviewer #1: No

Reviewer #2: Accept

**Summary and General Comments**

Reviewer #1: No comments

Reviewer #2: Suggested revisions have implemented, manuscript can be accepted

PLOS authors have the option to publish the peer review history of their article (what does this mean?). If published, this will include your full peer review and any attached files.

Reviewer #1: No

Reviewer #2: No

---

## [Editor Report · Acceptance letter]

10 Aug 2021

Dear MD Mischlinger,

We are delighted to inform you that your manuscript, "Diagnostic performance of capillary and venous blood samples in the detection of Loa loa and Mansonella perstans microfilaraemia using light microscopy," has been formally accepted for publication in PLOS Neglected Tropical Diseases.

Best regards,

Shaden Kamhawi

co-Editor-in-Chief

Paul Brindley

co-Editor-in-Chief
